# Aggression Unleashed: Neural Circuits from Scent to Brain

**DOI:** 10.3390/brainsci14080794

**Published:** 2024-08-08

**Authors:** Rhea Singh, Kyle Gobrogge

**Affiliations:** 1The Mortimer B. Zuckerman Mind Brain Behavior Institute, Department of Neuroscience, Columbia University, New York, NY 10027, USA; 2Undergraduate Program in Neuroscience, Boston University, Boston, MA 02215, USA; gobrogge@bu.edu

**Keywords:** aggression, rodents, ventromedial hypothalamus (VMHvl), medial amygdala (MeA), ventral premammillary nucleus (PMv), prefrontal cortex

## Abstract

Aggression is a fundamental behavior with essential roles in dominance assertion, resource acquisition, and self-defense across the animal kingdom. However, dysregulation of the aggression circuitry can have severe consequences in humans, leading to economic, emotional, and societal burdens. Previous inconsistencies in aggression research have been due to limitations in techniques for studying these neurons at a high spatial resolution, resulting in an incomplete understanding of the neural mechanisms underlying aggression. Recent advancements in optogenetics, pharmacogenetics, single-cell RNA sequencing, and in vivo electrophysiology have provided new insights into this complex circuitry. This review aims to explore the aggression-provoking stimuli and their detection in rodents, particularly through the olfactory systems. Additionally, we will examine the core regions associated with aggression, their interactions, and their connection with the prefrontal cortex. We will also discuss the significance of top-down cognitive control systems in regulating atypical expressions of aggressive behavior. While the focus will primarily be on rodent circuitry, we will briefly touch upon the modulation of aggression in humans through the prefrontal cortex and discuss emerging therapeutic interventions that may benefit individuals with aggression disorders. This comprehensive understanding of the neural substrates of aggression will pave the way for the development of novel therapeutic strategies and clinical interventions. This approach contrasts with the broader perspective on neural mechanisms of aggression across species, aiming for a more focused analysis of specific pathways and their implications for therapeutic interventions.

## 1. Introduction

Aggression is one of the oldest behavioral traits from an evolutionary perspective, serving as a mechanism to assert dominance, establish social hierarchies, secure resources, and defend against rivals. It is essential for individual and species survival, yet in humans, it can lead to extreme violence with severe outcomes such as death and societal distress. The innate versus learned nature of aggression is a subject of scientific inquiry [1]. Despite its negative impact on society, the neurobiology of aggression remains elusive, hindering the development of effective treatments. This review aims to explore the detailed neural pathways of aggression in rodents, emphasizing the olfactory system’s role in triggering aggression and comparing these mechanisms to human aggression.

Aggression can be triggered by various factors, including circadian rhythms, stress, reproductive status, and experiences of winning or losing [2]. In humans, aggression is categorized into reactive and instrumental subtypes. Reactive aggression, an impulsive response to provocation, is linked to hypothalamic and limbic systems. Instrumental aggression, which is premeditated harm, is associated with higher cortical functions [3]. Recent studies have also highlighted the association between trait aggression and five-factor personality traits in males. Specifically, higher aggression scores are positively associated with Neuroticism and negatively associated with Agreeableness and Conscientiousness [4]. These findings suggest that individuals high in Neuroticism and low in Agreeableness and Conscientiousness are at higher risk of exhibiting aggressive behavior, underlining the relevance of personality traits in understanding aggressive behavior.

Animals, particularly rodents, typically exhibit reactive aggression through hierarchically organized sensory transduction: smell, sight, and sound. The perception of an aggression-inducing signal can activate internal imbalances like hunger or disrupted circadian rhythms, altering aggression pathways. Research has shown that aggression and violence have their roots in neurobiology, involving key roles for neurotransmitters such as serotonin, norepinephrine, dopamine, glutamate, and γ-aminobutyric acid (GABA) [5]. Maladaptive aggression is influenced by neural activity in specific pathways and receptors within the serotonin system, which is modulated by catecholamines, GABA, and glutamate. Additionally, these neurotransmitter systems are modulated by neuropeptides, including vasopressin, oxytocin, corticotropin-releasing factor, and endogenous opioid peptides. These modulatory systems form intricate networks with several nodes that offer the potential for diagnostic measures and therapeutic interventions in both veterinary and human medicine [6].

This review focuses on reactive aggression in animals, its neural foundations, and the role of the prefrontal cortex in controlling instrumental aggression. We aim to bridge animal aggression studies with human applications, understanding the neural underpinnings of sudden uncontrollable aggression to inform interventions and policies within criminal justice and public health frameworks [7].

## 2. Methods

To inform our review, the following keywords in various combinations were entered into MEDLINE/PubMed, PsychINFO, and EMBASE databases in August 2024: aggression, violence, rodent aggression behavior, ventromedial hypothalamus, medial amygdala, ventral premammillary nucleus, core aggression circuit, rats, prefrontal cortex, human violence, and aggression-provoking stimulus. We limited our search to articles published in English. In addition, the authors performed further secondary screening and searches by using the references of all eligible papers. All titles and abstracts were examined closely, and full texts of potentially relevant papers were obtained to develop a representation of all the available literature on the selected topic. From an initial list of 181 studies, 78 were identified as most relevant for the current review.

## 3. Results and Discussion

### 3.1. Neural Substrates and Pathways of Aggression

#### 3.1.1. Aggression-Provoking Stimulus and Its Detection

The olfactory system plays a crucial role in rodent aggression, enabling them to recognize conspecifics and potential threats. Rodents, such as mice, possess specialized olfactory receptors that respond to specific scents like male urine or predator excretions, which trigger innate fear and aggression without prior learning [8]. There are two olfactory pathways: the main olfactory epithelium (MOE) and the vomeronasal organ (VNO). The MOE projects to the main olfactory bulb (MOB) and then to regions like the piriform cortex and cortical amygdala, which ultimately connect to the medial amygdala, which is critical for aggression onset [9]. The VNO, responsible for innate behavior responses, sends signals to the accessory olfactory bulb (AOB) and then to the medial amygdala [9] (Figure 1). 

Research has established that the olfactory system plays a critical role in initiating aggressive behavior. Studies have shown that mice with impaired sensory detection in either the main olfactory epithelium (MOE) or the vomeronasal organ (VNO) exhibit a marked reduction in aggression. For instance, the work by Dulac’s team revealed that mice deficient in the protein TRP2, which is found in VNO neurons, do not engage in inter-male aggression, underscoring the importance of sensory input from these neurons in sex discrimination [10]. Furthermore, TRP2^−/−^ male mice even initiated mating behavior towards other male mice, potentially indicating the TRP2^−/−^ males may be unable to distinguish males from females and thus engage indiscriminately in sexual behavior with a conspecific of either gender. In a related vein, mice lacking the functional cyclic nucleotide-gated channel alpha 2 (CNGA2) in the neurons of their MOE were less aggressive and showed diminished sexual interest in females [11]. These findings suggest the presence of an innate neural circuit in rodents that is responsible for detecting specific cues that trigger aggression, thereby increasing the propensity for aggressive behavior [2]. 

Further exploration reveals the roles played by the ventrolateral part of the ventromedial hypothalamus (VMHvl), the medial amygdala (MeA), and the ventral premammillary nucleus (PMV) within this aggression circuit, each of which are closely connected to influence aggressive behavior, as visualized in Figure 1.

#### 3.1.2. The Ventrolateral Portion of the Ventromedial Hypothalamus (VMHvl)

The Ventrolateral Portion of the Ventromedial Hypothalamus (VMHvl) is a key regulator of aggressive behaviors, including inter-male aggression and aggression-seeking actions. The significance of the VMHvl was first highlighted in a seminal study by Bard [12], which demonstrated that hypothalamic lesions, particularly within the lateral and ventromedial regions, diminish aggressive behaviors, underscoring the importance of these areas in the manifestation of aggression [12]. However, the precision of lesion studies is limited, as they affect both local cells and axonal fibers passing through the area.

Later, a study by Kruk employed electrical stimulation of the rat hypothalamus, pinpointing several regions, including the perifornical, anterior, lateral, and ventromedial hypothalamus, as being involved in aggression [13]. The advent of more refined techniques, such as optogenetics, has since allowed for precise targeting and understanding of these neuronal circuits. Groundbreaking work by Anderson and colleagues has shed significant light on the VMHvl’s role in aggression. Their chronic single-cell recordings in mice demonstrated that neurons within the VMHvl increase their firing rate during male–male encounters, specifically during aggressive attacks, suggesting that a distinct subset of neurons within the VMHvl are activated exclusively during aggressive behaviors [14]. Additionally, simultaneous recordings from 13 limbic regions in the social behavior network (SBN) revealed the VMHvl as the region with the largest and fastest activity increase during attack onset, highlighting the critical role this region plays in aggression [15].

Further research has uncovered that VMHvl neurons not only become active upon the introduction of an intruder but also maintain heightened activity levels after the intruder is removed. This prolonged activity suggests a sustained influence on the likelihood and timing of aggressive responses toward subsequent intruders [16,17]. Falkner identified that ‘male-responsive’ neurons within the VMHvl respond to proximity and attack velocity, with their peak activity occurring when an intruder is at close range, and the attack is delivered with high velocity [17].

Recent studies provide molecular insights into the role of oxytocin receptors (OXTRs) in aggression. Osakada et al. [18] demonstrated that OXTR-expressing neurons in the VMHvl are crucial for social avoidance following defeat in aggressive encounters. Pain during defeat activates VMHvlOXTR cells, leading to long-term potentiation in aggressor-related synapses and driving avoidance behavior to prevent future defeats, underscoring the role of OXTR signaling in defeat-induced social avoidance and fear learning. While not specific to VMHvl, Berendzen et al. [19] showed that brain-wide genetic modification of OXTRs in prairie voles did not impair social attachments or aggression towards unfamiliar conspecifics. OXTR-null voles still exhibited pair-bonding and aggression towards strangers, challenging the view of OXTR’s necessity in sociality. Together, these studies highlight the complexity and context-specific roles of OXTR signaling in modulating aggression.

Research has also expanded our understanding of aggression to include steroid-specific behaviors, focusing on VMHvl neurons that express receptors for progesterone and estrogen. These neurons regulate social interactions in both sexes [20]. A breakthrough study using fluorescent markers in female mice demonstrated that these neurons are highly active during episodes of aggression and sexual behaviors, marking the first such observation in female mice [21]. Nair reported that most VMHvlEsr1 neurons exhibit mixed behavioral selectivity in both imaging and transcriptomic studies, suggesting that the VMHvl represents behavior via population coding and not cell identity [22].

Additionally, aggression-regulating pathways have been identified that can be activated or inactivated to suppress or enhance aggressive behavior. Esr1-expressing cells in the caudal part of the medial preoptic area (cMPOA) project to the VMHvl, implicating this pathway in modulating intermale aggression bidirectionally [23]. The study found that cMPOAEsr1 cells also encode information such as the resource-holding potential of opponents. Using fiber photometry recordings of cMPOAEsr1 cells, a negative correlation was discovered between cMPOAEsr1 cell activity and time spent investigating or engaging with an animal. Chemogenetic inhibition of cMPOAEsr1 cells enhanced aggression in naturally aggressive animals but did not induce attack behavior in nonaggressive animals. Furthermore, cMPOAEsr1 cells regulate aggression by inhibiting VMHvlEsr1 cells, and activation of the cMPOAEsr1–VMHvl pathway is shown to suppress aggression [23].

The sensitivity of VMHvlPR neurons has also been further substantiated by demonstrating their activity during aggressive encounters occurring between other mice. Fiber photometry imaging of VMHvlPR neurons while animals engaged in attack behavior, as well as while an animal could only witness attack behavior, revealed an overlap in neuron activity, suggesting that VMHvlPR neurons are active when a mouse is fighting and when it witnesses fighting between others [24]. In 2022, research further delineated the role of the VMHvl in female aggression. Through single-cell RNA sequencing, the study identified distinct transcriptomic cell types in the VMHvl of females and observed variations in aggressive behavior across different stages of lactation [25]. By selectively inhibiting certain VMHvl cells, researchers noted a decrease in maternal aggression. Conversely, stimulating these cells provoked aggressive behavior, highlighting their integral role in modulating aggression in females [25].

The sexual dimorphism of the aggression circuitry in rodents is another interesting avenue of research. Recently, a study demonstrated a hypothalamic–amygdala circuit that underlies sexually dimorphic aggression. The study highlighted circuitry of the VMHvl that projects to the posterior substantia innominata (pSI) and promotes attack in both sexes of mice. The findings revealed a sexually distinct excitation–inhibition balance of the circuit; excitatory VMHvl-pSI projections are strengthened in males to promote aggression, whereas the inhibitory connections that reduce aggressive behavior are strengthened in females [26]. The study has key implications for how defined changes in this excitatory–inhibitory balance may affect social interactions [26].

Additionally, it is important to note the flexibility of VMHvl cell responses [27]. For instance, male and female-induced activation patterns in the VMHvl become more distinct with sexual experience, whereas inexperienced animals activate overlapping neuronal populations [28]. Another study emphasized the role of experience in modulating VMHvl activity by showing that aggression experience and consecutive winning experiences in mice led to increased long-term potentiation of excitatory synaptic inputs in the VMHvl [29]. The significance of hypothalamic modulation of aggression extends beyond mice and has been observed in other rodent species, including gerbils, hamsters, and voles, though it appears more anterior-driven in these species [30,31,32].

Recent studies have also led to a view that challenges behavioral control via the hypothalamus. This breakthrough study asserts that distinct but anatomically distributed neuron ensembles encode the social and fear behavior classes through mixed selectivity [33]. This concept underscores the idea that behavior representation in the hypothalamus is not governed by a single dedicated set of neurons but involves distributed and versatile neural ensembles, thus challenging the view that the encoding of behavior is restricted to a single nucleus or region [33]. Additionally, a one-of-a-kind study into the VMHvl attempted to link transcriptomic diversity in the CNS to behavior function. The study identified 17 transcriptomic neuron types (T-types) using single-cell RNA sequencing. Despite finding some sexually dimorphic clusters and neurons with different projection target preferences, the analysis revealed that few VMHvl T-types showed clear behavior-specific activation, indicating a complex relationship between cell type, connectivity, and social behaviors like aggression and mounting [34].

The current research highlights the intricacies of hypothalamic pathways in regulating aggression, considering factors such as sexual dimorphism and the distinct types of cells within the VMHvl. While it might seem straightforward to view the hypothalamus as the central command for aggression control, such a simplistic view fails to capture the complex nature of aggressive responses as delineated by Falkner and Lin [35]. The intricate interplay among the hypothalamus, amygdala, and premammillary nucleus in the context of aggression offers a promising area for future studies aimed at elucidating the elaborate network that underlies aggressive behavior [17].

#### 3.1.3. The Medial Amygdala (MeA)

The medial amygdala (MeA) plays an integral role in processing sensory information and triggering aggressive behaviors [10,36,37]. Specifically, the posteroventral division of the MeA (MePV) is a critical juncture for excitatory signals from both the accessory and main olfactory systems, integrating olfactory cues [9]. Amygdala lesions in rats have been shown to suppress aggressive behaviors, with the extent of aggression reduction directly proportional to the lesion size, emphasizing the MeA’s significance in modulating emotional valence. Consistently, MeA activation during male aggressive encounters is evidenced by the expression of the neuronal activation marker c-Fos across various rodent species [38,39].

In prairie voles, post-mating males exhibit increased aggression towards intruders alongside a rise in Fos-immunoreactive cells in the MeA, irrespective of the intruder’s sex [40]. This response is corroborated in pair-bonded males who show aggression towards unfamiliar males and females alike [30], suggesting a robust association between MeA activation and aggressive behavior across rodent species. 

Various research techniques, including electrophysiology, lesioning, and optogenetics, have been utilized to dissect the MeA’s function in aggression. Anderson’s team highlighted that the posterior dorsal subdivision of the MeA (MeApd) expresses c-Fos during aggressive acts. Optogenetic activation of neurons in the MeApd precipitated attack behaviors in males, whereas inhibiting GABAergic neurons in the MeApd curbed aggression entirely, indicating the MeApd’s pivotal role in such behaviors [41].

Distinct response patterns and functions have also been observed in MeA cells originating from two embryonically parcellated developmental lineages: MeAFoxp2 and MeADbx1. Fiber photometry imaging revealed that MeAFoxp2 cells showed male-specific responses slightly after stimulus onset, with activity levels increasing significantly during attack initiation. In contrast, MeADbx1 cells responded to all social stimuli immediately at stimulus onset, indicating more sensory-related activity cues rather than action phases of behavior [42].

A study from 2024 identified a specific neural circuit involving tachykinin-expressing neurons in the medial amygdala (MeATac1 neurons). These neurons are activated during aggression and project to the VMHvl. Substance P (SP) and its preferred neurokinin-1 receptor (NK-1R) play multifaceted roles in modulating behavior across the nervous system. Using techniques such as fiber photometry, chemogenetic manipulations, and patch-clamp recordings, researchers found that SP/NK-1R signaling modulates excitatory transmission. Activation of these neurons using DREADDs (Designer Receptors Exclusively Activated by Designer Drugs) elicited attacks against female intruders in male mice, while activation of MeA → VMHvl neuron activity reduced attack latency and increased attack duration towards male conspecifics [43]. These findings suggest that SP/NK-1R signaling is integral to aggression control and may offer therapeutic potential for managing aggression-related behaviors [43].

Aggression priming, where previous aggressive episodes heighten future aggression, has been linked to the activation of the posterior ventral segment of the MeA (MeApv). The MeApv, connected to both the ventromedial hypothalamus (VMH) and the bed nucleus of the stria terminalis (BNST), influences the duration of attacks [44]. Importantly, the aggression-regulating circuitry of the MeA is not exclusive to males; females exhibit similar patterns. For instance, increased c-Fos expression in the MeA of aggressive female Syrian hamsters parallels the response seen in males [45]. In meadow voles, reducing estrogen receptor alpha (ERα) in the MeA decreased aggression and promoted prosocial behaviors. Conversely, modifying ERα levels in the VMH influenced mating behaviors and amplified same-sex aggression [46].

The MeA’s role extends to maternal aggression as well. Lactating mice display heightened MeA activity in the presence of male intruders, an effect not observed in virgin females, potentially reflecting an aversive reaction to male pheromones and the MeA’s olfactory connectivity [47]. While inhibiting the MeA does not completely abolish maternal aggression, it affects the escalation of such behaviors [48], highlighting the MeA’s involvement in modulating aggression intensity and persistence [48].

#### 3.1.4. Ventral Premammillary Nucleus (PMv)

The ventral premammillary nucleus (PMv), situated posterior to the VMHvl, plays a crucial role in a variety of behaviors, including maternal aggression, mating, exploratory actions, and intermale aggression [33]. It receives and processes information from various brain regions, such as the medial nucleus of the amygdala, the bed nucleus of the stria terminalis, and the medial preoptic nucleus. The PMv’s responsiveness to scents from the opposite sex underscores its integral position in the neural circuitry governing aggression [49].

In female rats, specific bilateral lesions in the PMv have been shown to significantly diminish maternal aggression while leaving maternal care and investigative behaviors intact. This suggests that the PMv is particularly crucial for the perception of potential threats from intruders. Additionally, these lesions lead to a decrease in Fos expression within the VMHvl, implying that VMHvl activation is dependent on input from the PMv, thereby revealing the circuitry’s complex interconnectedness and its functionality in triggering context-specific forms of aggression [50].

The PMv also has a significant role in mediating intermale aggression. Neurons within the PMv are activated in the presence of a male intruder but remain inactive in response to females, which indicates a male-specific mechanism for triggering aggression [51]. Moreover, PMv neurons that express the dopamine transporter (PMvDAT neurons) are key in driving goal-oriented aggressive behaviors in male mice, correlating with aggression intensity and the establishment of social hierarchies [52].

Using c-Fos as a marker for neuronal activation, studies have observed greater activity in aggressive males compared to non-aggressive counterparts, with a positive correlation between the duration of attacks and the number of activated c-Fos+DAT+ neurons in the PMv. This heightened discharge of PMvDAT neurons in aggressive males suggests a strong association with intermale aggression. Interestingly, low-intensity optogenetic stimulation of PMvDAT neurons induces investigative behaviors in aggressors, while higher intensity provokes attacks. In contrast, non-aggressive males only exhibit investigative responses to the same stimulation and do not progress to aggression. Additionally, modulating the activity of PMvDAT neurons can influence the hierarchical status within a social group [52].

The fact that PMv stimulation does not elicit aggression in non-aggressive individuals prompts further investigation into the social hierarchy’s influence on long-term aggression responses. This insight opens avenues for additional research to delve into the complex dynamics of aggression and the precise role of the PMv in these processes [52].

Our examination of the VMHvl, MeA, and PMV underscores the intricate integration of these structures in the orchestration and modulation of aggression. Research has shown that the activation or inhibition of these regions can significantly alter aggressive behaviors in a time-sensitive manner [2]. The table below highlights a visual summary of the neural substrates of aggression discussed, indicating the increase or decrease of neuronal activity in the specified brain region based on the targeting mechanism (Table 1). Despite the advances in aggression research over the past decade, there remains a considerable gap in our understanding. The studies discussed indicate that targeting these regions broadly is insufficient for eliciting specific aggressive responses or behaviors. Future research will benefit from techniques like single-cell RNA sequencing (scRNA-seq), which can identify the precise neurons responsible for various forms of aggression within different social contexts [2].

Furthermore, distinguishing between the manifestation of aggressive behaviors and an aggressive internal state is a critical aspect that warrants deeper investigation [2]. Additionally, while the VMHvl, MeA, and PMV are pivotal in aggression regulation, they do not operate in isolation. Given the high cost of aggression, it is subject to strict regulation by superior neural mechanisms. Historical experiments on cats as far back as 1928 demonstrated that disconnecting the forebrain from posterior structures, leaving only the hypothalamus and its downstream pathways intact, resulted in spontaneous and unprovoked rage [53]. This finding points to the significant influence of upper-tier structures, such as the prefrontal cortex (PFC), in aggression control [53].
brainsci-14-00794-t001_Table 1Table 1Aggression Neural Substrates.AnimalBrain TargetReferencesNeuronal Directionality *Molecular PhenotypeMouseVMHvlLin, 2011 [14]↑N/A **

Yang, 2013 [54]↑PR+/Esr1+

Falkner, 2014 [17]↑N/A

Lee, 2014 [20]↑Esr1+

Hashikawa, 2017 [21]↑PR+/Esr1+

Liu, 2022 [25]↑Npy2r+ (β) cells

Guo, 2023 [15]↑N/A

Nair, 2023 [22]↑Esr1+

Wei, 2023 [23]↓cMPOA-VMHvl Esr1+

Yang, 2023 [24]
PR+MouseMeAWang, 1997 [40]↑N/A

Dulac et al., 2003 [10]↓Trp2+

Hasen Gammie et al., 2005 [47]↑N/A

Hong et al., 2014 [41]↑GABA

Nordman, 2020 [44]↑NMDAR

Abellán-Álvaro et al., 2022 [48]↓N/A

Lischinsky, 2023 [42]↑Foxp2+

He, 2024 [43]↑Tac1+MousePMvSoden, 2016 [51]↑PMv-DAT

Stagkourakis et al., 2018 [52]↑PMv-DATRatVMHvlBard, 1958 [12]↓N/A

Kruk, 1983 [13]↑N/A

Veening, 2005 [39]↑N/ARatMeAVochteloo Koolhaas, 1987 [55]↓N/A

Veening, 2005 [39]↑N/ARatPMvMotta, 2013 [50]↑N/A

Cavalcante, 2014 [49]↓N/ASyrian HamsterVMHvlKollack-Walker et al., 1997 [56]↑N/A

Delville et al., 2000 [57]↑N/A

Pan et al., 2010 [58]↑N/ASyrian HamsterMeAPotegal et al., 1996a [59]↑N/A

Kollack-Walker et al., 1995 [60]↑N/APrairie VoleAHGobrogge et al., 2007 [30]↑TH/AVP

Gobrogge et al., 2009 [61]↑AVP (V1aR)

Gobrogge et al., 2016 [62]↑AVP/CRH


↓5-HTPrairie VoleMeAWang et al., 1997 [40]↑N/A

Stetzik et al., 2018 [46]↓Esr1+Meadow VoleMeAPan et al., 2019↑N/AMongolian GerbilVMHvlPan et al., 2020 [31,32]↑N/A* Note: The arrows in the aforementioned column indicate an increase or decrease of neuronal activity in the specified brain region associated with the expression of aggressive behavior across species. ** N/A indicates that the study did not target a particular molecular cell-type but rather looks at neuronal activity in the region as a whole.


The subsequent sections of this paper will delve into current research on the PFC, exploring its role in managing aggression and how its dysregulation can lead to excessive instrumental aggression. This is not only pertinent to our understanding of animal behavior but also has profound implications for human neuropsychiatry.

### 3.2. The Prefrontal Cortex and Its Control Over Aggression

The prefrontal cortex (PFC) is known to significantly influence aggressive behavior. In rodents, regions such as the medial prefrontal cortex (mPFC), orbital prefrontal cortex (OFC), and dorsolateral prefrontal cortex (DLPFC) have been implicated in the modulation of intermale aggression. Early research, dating back to 2006, identified strong c-Fos activation in the mPFC’s infralimbic and prelimbic subdivisions during aggressive interactions [63]. Moreover, social isolation seems to exacerbate aggressive behaviors, with isolation altering the activation of immediate early genes (IEGs) in a sex-dependent manner [64]. Socially isolated rats showed muted IEG responses to new conspecifics compared to group-housed rats, with isolated males displaying increased social interaction and both sexes showing more aggressive grooming, behaviors associated with heightened stress [64].

A 2023 study demonstrated that post-weaning social isolation (PWSI) induced abnormal aggressive behavior [65]. Resident-intruder tests showed that PWSI mice exhibited enhanced aggression compared to socially reared mice, with shorter attack latencies, increased frequency of offensive behavior, and more violent bites to vulnerable body parts compared to social mice. Upon conducting c-Fos immunostaining on the mouse brains 90 min after an aggressive encounter, it was found that the aggressive interaction increased c-Fos expression in the OFC and mPFC compared to baseline-resting groups [65]. Interestingly, the aggressive encounter induced neuronal hyperactivation in the mPFC of PWSI mice, as well as disrupted co-activation patterns in the PFC [65]. These results demonstrate how early-life social stress and isolation can impact the maturing PFC and lead to the development of social abnormalities in adulthood.

Another study in Syrian female hamsters aimed to evaluate the intracellular signaling mechanisms that control the neurophysiological plasticity that underlies the rewarding consequences of aggressive interactions [66]. The hamsters were made to engage in up to five aggressive encounters with a male conspecific, after which the phosphorylation of specific proteins in the forebrain was studied. The results of the study depicted a decrease in the phosphorylation of mTOR in the medial prefrontal cortex 10 min following the second aggressive interaction [66]. The authors suggest that this may imply translational events that induce long-lasting synaptic changes, as previous studies have implicated increased mTOR phosphorylation in mice to promote social behavior [67].

These findings suggest that social isolation impacts the mPFC’s response to social stimuli, with potential implications for understanding social deficits in humans. In particular, the negative correlation between Arc and c-Fos in the mPFC during aggressive grooming in male rats indicates a possible reliance on PFC–limbic interactions that may differ from females.

The inhibitory role of the mPFC on aggression was further elucidated by studies employing optogenetics to manipulate neuronal activation. Activation of excitatory neurons in the mPFC during a resident-intruder assay was shown to decrease aggression while silencing these neurons increased it [68]. Notably, changes in mPFC activity did not affect the duration of aggressive episodes, suggesting that while the mPFC can modulate the initiation of aggression, it cannot curtail an ongoing aggressive event [68]. Contrarily, OFC activation did not affect aggression, indicating a need for more targeted research to understand its role [68].

Research into rodent aggression systems may hold key implications down the line for human aggression. While the PFC control of social behaviors is significantly more complex in humans than in rodents, research into the mPFC of rodents has allowed us to expand our understanding of the possible role of the prefrontal cortex in regulating social behaviors and how disorders in these regions can exacerbate violent tendencies and aggression [68]. The discussion will now briefly touch upon the human mPFC and its regulatory role in aggression so as to give a high-level overview of some of the brain regions in the human PFC that affect behavior. Historical and contemporary case studies, including the well-known incident of Phineas Gage and subsequent neuroimaging research, have linked mPFC damage with disruptions in social and moral behaviors and an increased propensity for aggression [69,70]. Similar patterns of behavior and neural dysfunction have been observed in patients with Intermittent Explosive Disorder (IED), suggesting a commonality in the dysfunction of inhibitory projections from the mPFC to the amygdala, resulting in a failure to modulate aggressive responses [71,72].

Recent studies have explored the DLPFC’s involvement in aggression through interventions like transcranial direct current stimulation (tDCS), which has shown promise in reducing aggressive intentions [73]. These findings support the hypothesis that enhancing PFC activity can lead to reduced impulsivity and aggression [73].

Furthermore, research in populations of aggressive, violent offenders has highlighted changes in amygdala connectivity with the mPFC and paralimbic regions during anger provocation, suggesting a mechanism where heightened emotional experiences necessitate increased regulatory efforts from the PFC and, when overwhelmed, may result in violent outbursts [74,75].

In conclusion, while the amygdala is key in the generation of aggression, the PFC, particularly the mPFC, plays a vital role in regulating these behaviors. Future research should focus on the observed changes in the amygdala–PFC connectivity and their relation to aggressive behavior severity, exploring interventions that could modify this connectivity as potential treatments for aggression-related disorders.

## 4. Future Directions and Concluding Remarks

The field of aggression research has advanced rapidly over the past decade, invigorated by the advent of genetic technologies that enable the precise targeting of specific neuronal populations. This paper has examined the olfactory system’s role in detecting cues that provoke aggression and the sophisticated interplay among the ventromedial hypothalamus, medial amygdala, and ventral premammillary nucleus in eliciting aggressive responses. We have also highlighted the prefrontal cortex’s regulatory influence on aggression, with a particular focus on its capacity to curb impulsive and reactive tendencies. Despite these insights, the journey into the neural underpinnings of aggression is far from complete.

Future research should leverage cutting-edge methodologies like single-cell RNA sequencing (scRNA-seq) to unravel the neuronal diversity underpinning aggression’s varied manifestations. The pioneering studies by Dayu Lin and colleagues have set the stage for an in-depth exploration of the VMHvl, promising to pinpoint specific neurons that modulate aggression within diverse social contexts. A crucial area of investigation is the distinction between aggressive actions and internal states of aggression. Understanding how core aggression circuitry differentiates and regulates these states is vital for a comprehensive grasp of aggression’s multifaceted nature. Elucidating the pathways that transition an internal aggressive state to outward aggressive behaviors will be invaluable [34].

Bridging the translational gap between animal models and human aggression is another frontier that demands attention. Longitudinal studies that monitor the evolution of aggressive behaviors in individuals with prefrontal dysfunctions, coupled with assessments of neuromodulatory interventions like tDCS, are critical for developing effective treatments [73].

The neuromodulatory basis of aggression is also an avenue that is key in advancing our understanding of how the intensity of aggression is modulated in accordance with changes in the external environment [76]. The flexibility in outputs of the same behavior from a genetically hardwired circuit is noteworthy. A deeper understanding of the roles of serotonin, dopamine, oxytocin, estrogen, vasopressin, and other neuromodulators implicated in aggression is essential in identifying how learned experiences may lead to persistent changes in our circuitry [77].

Due to the small body mass of rodents, it has often been difficult to monitor blood serum hormone levels over time. The development of novel fluorescent indicators for neuromodulatory molecules opens the door to exciting future possibilities of tracking in vivo sex hormone dynamics across broad timescales [78]. Future tools that could potentially inactivate specific neuromodulator-releasing neuronal populations or knock out identified gene targets in specific cell populations would allow the testing of the temporal dynamics of neuromodulator release, as well as experience-dependent plasticity [78].

Moreover, considering that aggression manifests variably across species, a comparative approach could shed light on universal mechanisms and species-specific nuances of aggression, enhancing our understanding of human aggression’s complexities [42]. While it is clear that given the differences in biology and structure, humans and rodents cannot be classified into the same categories, many neurochemical systems have evolved to regulate species-specific aggression. While aggression in humans and rodents is expressed in significantly different contexts, manifestations, and behavioral outputs, there is a degree of overlap in neuroanatomical and neurochemical pathways [3]. There is a myriad of questions in relation to human aggression that require years more of investigation. How does violence in media affect the violent behavior of individuals? How does one’s socioeconomic background and availability of base resources such as food and shelter affect the development of these neuronal circuits? How does childhood trauma lead victims to continue to perpetrate aggressive behavior? These questions require a combination of both biology-based and survey-based investigations to allow precise quantitative measurements of brain activity during the engagement in violent behaviors, as well as a qualitative examination of how individuals perceive their behavior and their interactions with their surroundings.

The path forward in aggression research lies in the synthesis of physiological techniques and connectivity analyses alongside innovative genetic manipulation strategies. Targeting neurons based on their functional relevance, projection patterns, or molecular characteristics will be instrumental in catalyzing new discoveries in this dynamic domain of neuroscience.

In summary, while we have charted significant territories in the landscape of aggression research, the horizon teems with unexplored questions and potential. The continued effort to integrate diverse scientific disciplines and technological advancements will undoubtedly lead to deeper insights and more refined interventions to address the pervasive challenge of aggression in society.

## Figures and Tables

**Figure 1 brainsci-14-00794-f001:**
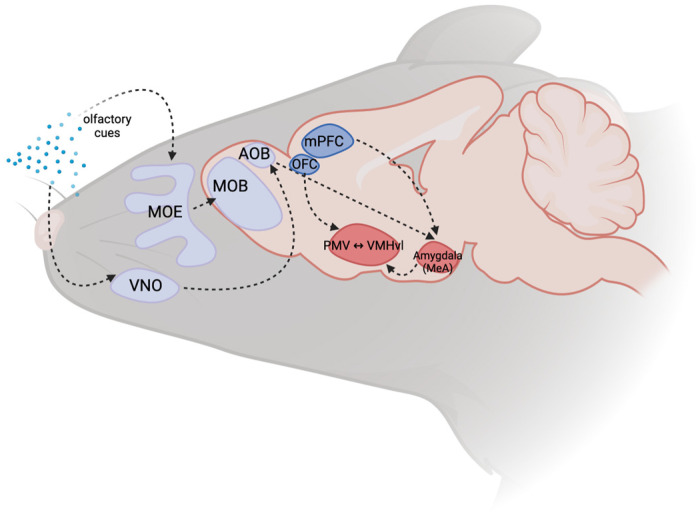
The Ventrolateral Portion of the Ventromedial Hypothalamus (VMHvl) (figure designed using BioRender Scientific Image and Illustration Software|BioRender. Available online: https://www.biorender.com/ (accessed on 15 June 2024).

## Data Availability

Not applicable.

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
