# Peer review of "Aggression Unleashed: Neural Circuits from Scent to Brain"

_brainsci, 2024, doi:10.3390/brainsci14080794_

Round 1

Reviewer 1 Report

Comments and Suggestions for Authors

Thank you for the opportunity to review this paper. The topic sounded interesting, but some methodological issues need to be addressed. 

I'm not quite sure about the study's aim. The Authors declare that they want to compare rodents with human beings, but most of the material is repetitive and doesn't provide any new knowledge.

There is a lack of clearly described methodology. How did the Authors decide on choosing the literature? Why did they skim other researchers' research?

I'm not sure if the term "recent" in neuroscience applies to the research from 2019 (line 310). 

There are not too many recent papers. Even the above-mentioned work of Siep et al. had more than 60 citations. Maybe the Authors should analyse the literature more insightfully.

To sum up, I suggest that the authors describe their methodological background better and focus on the newest literature in the field.

Author Response

Comment 1: I'm not quite sure about the study's aim. 

Response 1: Thank you for your feedback. We have clarified the aim of the study in the introduction, emphasizing the exploration of neural pathways of aggression in rodents and their comparison to human aggression.

Comment 2: There is a lack of clearly described methodology.

Thank you for pointing this out. We have added a detailed methodology section, explaining the criteria for literature selection and the databases used.

Comment 3: There are not too many recent papers. Even the above-mentioned work of Siep et al. had more than 60 citations. Maybe the Authors should analyse the literature more insightfully.

Thank you for your feedback. We have incorporated more recent papers, including ten additional sources from the past three years, to ensure our discussion is up-to-date.  We have expanded our discussion on recent studies and their implications for understanding aggression.

Reviewer 2 Report

Comments and Suggestions for Authors

Unraveling the Neural Pathways of Aggression in Rodents: From Scent to Strike

As a global suggestion, I advise modifying the writing registry, specifically avoiding expressions such as “Dr. David J. Anderson” (With all the genuine respect for Anderson) and limiting to report and describe the findings. Use please, Name et al., or “and colleagues”.

Introduction: Aggression from a psychological point of view is considered a dimensional construct (take a look to “Dam VH, Hjordt LV, da Cunha-Bang S, Sestoft D, Knudsen GM, Stenbaek DS. Trait aggression is associated with five-factor personality traits in males. Brain Behav. 2021 Jul;11(7):e02175. doi: 10.1002/brb3.2175. Epub 2021 May 25. PMID: 34036747; PMCID: PMC8323029”. Not mine and only advise)., implying its complexity. For this reason, I advise to remove or fix the first sentence(line 28 -30). Please, revise.

Line 35- I suggest revising the sentence, distinguishing in a better way the factors that can trigger aggressive behavior.

Line 36- In humans, aggression is 36 categorized into reactive and instrumental subtypes. This appears to be an oversimplification that needs to be overcome.

Line 46 - uncontrollable aggression to inform interventions and policies within 46 criminal justice and public health frameworks. This sentence needs to be revised since there is a jump. I agree that is interesting to review aggression in animals and related brain underpinnings, but the theoretical “Jump” appears to be evident and I advise avoiding this, explaining and stating the hypotheses.

Despite the narrative nature of the present review, a brief paragraph about the electronic search, the databases, and keywords is needed. This is usually performed also for narrative reviews, and you can find several previously published examples.

Discussion: This section appears ex abrupto, and it is quite confusing. I advise to name this section, differently and at the end writing a conclusion, in which future directions can be included.

The sentence between 64 – 68 needs to be revised and written differently. It is not clear if the mice TPR2 were -/- and what happens. Please describe better.

At the end of 2.1.1. you introduced the next sub-sections, which is preferable to motivate.

The Fig.1 is fine.

2.1.3. is well written, but, please,  take into consideration stylistic commentaries (as above).

However, at the end of each section/sub-section, I advise a summary of the content, instead than an introduction to the following one.

2.2. The Prefrontal Cortex and its Control Over Aggression- This section needs to be rewritten since the role of human PFC and its sub-regions is very complex. Moreover, I suppose that is quite off-topic since the principal purpose is “Rodent” aggressive behavior.

3. Future Directions and Concluding Remarks. In this section, a more comprehensive overview is needed.  

Author Response

Comment 1: As a global suggestion, I advise modifying the writing registry, specifically avoiding expressions such as “Dr. David J. Anderson” (With all the genuine respect for Anderson) and limiting to report and describe the findings. Use please, Name et al., or “and colleagues”.

Thank you for this feedback.  We have revised the writing style to use "Name et al." instead of full names, following the recommendation to avoid expressions such as “Dr. David J. Anderson.”

Comment 2:  Aggression from a psychological point of view is considered a dimensional construct (take a look to “Dam VH, Hjordt LV, da Cunha-Bang S, Sestoft D, Knudsen GM, Stenbaek DS. Trait aggression is associated with five-factor personality traits in males. Brain Behav. 2021 Jul;11(7):e02175. doi: 10.1002/brb3.2175. Epub 2021 May 25. PMID: 34036747; PMCID: PMC8323029”. Not mine and only advise)., implying its complexity. For this reason, I advise to remove or fix the first sentence(line 28 -30). Please, revise.

Thank you for your feedback.  We have included lines 28-30, but have made changes in the introduction. The introduction has been revised to better distinguish factors that trigger aggressive behavior and to incorporate the dimensional construct of aggression, referencing Dam et al. (2021).

Comment 3: Line 35- I suggest revising the sentence, distinguishing in a better way the factors that can trigger aggressive behavior.

We have corrected the sentence to "Aggression can be triggered by various factors, including circadian rhythms, stress, reproductive status, and experiences of winning or losing".

Comment 4: In humans, aggression is 36 categorized into reactive and instrumental subtypes. This appears to be an oversimplification that needs to be overcome.

Thank you for this comment. We have corrected the sentence to "In humans, aggression is categorized into reactive and instrumental subtypes. Reactive aggression, an impulsive response to provocation, is linked to hypothalamic and limbic systems."

Comment 5: uncontrollable aggression to inform interventions and policies within 46 criminal justice and public health frameworks. This sentence needs to be revised since there is a jump. I agree that is interesting to review aggression in animals and related brain underpinnings, but the theoretical “Jump” appears to be evident and I advise avoiding this, explaining and stating the hypotheses.

Thank you for this feedback. We have corrected the sentence to "We aim to bridge animal aggression studies with human applications, to better inform our understanding of the neural underpinnings of sudden, uncontrollable aggression. Aggression research holds far-reaching implications for interventions and rehabilitation down the line, so as to inform interventions and policies within criminal justice and public health frameworks."

This allows us to distinguish between our work of reviewing the current aggression research and its connections to human applications, while also mentioning the theoretical and eventual implications aggression research holds in the future of neuroscience.

Comment 6:  Despite the narrative nature of the present review, a brief paragraph about the electronic search, the databases, and keywords is needed. This is usually performed also for narrative reviews, and you can find several previously published examples.

Thank you for pointing this out. We have added a comprehensive methods section detailing our electronic search strategy, databases, and keywords used.

Comment 7: Discussion: This section appears ex abrupto, and it is quite confusing. I advise to name this section, differently and at the end writing a conclusion, in which future directions can be included.

Thank you for your feedback. We have corrected this by removing the section title of discussion, and renaming it to our title heading, and have made the final sections the conclusion and future directions. 

Comment 8:  The sentence between 64 – 68 needs to be revised and written differently. It is not clear if the mice TPR2 were -/- and what happens. Please describe better.

Thank you for your feedback. To clarify this further, we have corrected this to "Furthermore, TRP2 -/- male mice even initiated mating behavior towards other male mice, potentially indicating the TRP2−/− males may be unable to distinguish males from females and thus engage indiscriminately in sexual behavior with a conspecific of either gender.".

Comment 9: However, at the end of each section/sub-section, I advise a summary of the content, instead than an introduction to the following one.

Thank you for your feedback. We have taken it into account and edited the end of sub sections accordingly.

Comment 10: The Prefrontal Cortex and its Control Over Aggression- This section needs to be rewritten since the role of human PFC and its sub-regions is very complex. Moreover, I suppose that is quite off-topic since the principal purpose is “Rodent” aggressive behavior.  Future Directions and Concluding Remarks. In this section, a more comprehensive overview is needed.  

Thank you for your insights. To further strengthen our article, we have done the following: 

    • Added brief concluding paragraphs to the Aggression Provoking Stimulus and its Detection, and MeA sections to summarize content.
    • Expanded the discussion on the role of the prefrontal cortex (PFC) in aggression, emphasizing its regulatory role and implications for human behavior.

Round 2

Reviewer 1 Report

Comments and Suggestions for Authors

Thank you for applying my suggestion.

Reviewer 2 Report

Comments and Suggestions for Authors

The authors addressed my concerns